# Linking Expansion Behaviour of Extruded Potato Starch/Rapeseed Press Cake Blends to Rheological and Technofunctional Properties

**DOI:** 10.3390/polym13020215

**Published:** 2021-01-09

**Authors:** Anna Martin, Raffael Osen, Heike Petra Karbstein, M. Azad Emin

**Affiliations:** 1Department of Food Process Development, Fraunhofer Institute for Process Engineering and Packaging IVV, 85354 Freising, Germany; raffael.osen@ivv.fraunhofer.de; 2Food Process Engineering, Institute of Process Engineering in Life Sciences, Karlsruhe Institute of Technology, 76131 Karlsruhe, Germany; heike.karbstein@kit.edu (H.P.K.); azad.emin@kit.edu (M.A.E.)

**Keywords:** extrusion, plant proteins, canola, expansion, rheology

## Abstract

In order to valorise food by-products into healthy and sustainable products, extrusion technology can be used. Thereby, a high expansion rate is often a targeted product property. Rapeseed press cake (RPC) is a protein- and fibre-rich side product of oil pressing. Although there is detailed knowledge about the expansion mechanism of starch, only a few studies describe the influence of press cake addition on the expansion and the physical quality of the extruded products. This study assessed the effect of RPC inclusion on the physical and technofunctional properties of starch-containing directly expanded products. The effect of starch type (native and waxy), RPC level (10, 40, 70 g/100 g), extrusion moisture content (24, 29 g/100 g) and barrel temperature (20–140 °C) on expansion, hardness, water absorption, and solubility of the extrudates and extruder response was evaluated. At temperatures above 120 °C, 70 g/100 g of RPC increased the sectional and volumetric expansion of extrudates, irrespective of starch type. Since expansion correlates with the rheological properties of the melt, RPC and RPC/starch blends were investigated pre- and postextrusion in a closed cavity rheometer at extrusion-like conditions. It was shown that with increasing RPC level the complex viscosity |ƞ*| of extruded starch/RPC blends increased, which could be linked to expansion behaviour.

## 1. Introduction

The cultivation and processing of rapeseed—primarily for the production of rapeseed oil—increased worldwide from 50.6 mt in 2007 to 76.2 mt in 2017 [1]. About 50 mt of rapeseed press cake (RPC) accumulates as a residue every year. RPC comprises the solid residues that remain after the extraction of oil from the oil containing plant parts such as the seeds. Depending on the extraction method, the protein, lipid and fibre content of RPC can vary largely and lies in the range of 19–40 g/100 g, 5–18 g/100 and 11–14 g/100 g [2,3,4,5,6,7]. The amino acid profile of RPC has been shown to be comparable to soybean press cake but is richer in sulphur-containing amino acids, e.g., methionine and cysteine [8]. In addition, RPC has a higher fat content than soybean press cake and is a good source of calcium, iron, manganese, selenium and many B vitamins. It is also one of the richest sources of nonphytate phosphorus. Due to its all-year availability, competitive price and high protein content, RPC has received attention as a possible feed component for livestock and aquaculture [9,10]. 

Up until now there has only been limited use of RPC. This has mostly been for animal and not human nutrition due to the fact that RPC contains high contents of antinutritional factors (ANFs) such as phenolic compounds, phytic acid and glucosinolates [11,12,13]. However, various processing methods have been studied that successfully reduce the ANF content of RPC (i.e., dehulling, steaming, toasting, wet heating, water washing, chemical treatment, microbial degradation [14], seed germination [15,16], enzymatic treatment and fermentation [6,17]). This opens up the possibility to valorise RPC into, i.e., directly expanded snacks in order to produce sustainable new food products with an enhanced nutritional value. 

Low moisture extrusion is one technology used for texturising starch-based and protein-based biopolymers into food products having an expanded porous structure. There is detailed knowledge about the expansion mechanism of several starch types, but only two studies describe the influence of oilseed press cake addition on the expansion and the physical quality of the extruded products. To our knowledge, there are no studies investigating the impact of RPC on food products. For instance, it has been reported recently that the addition of RPC to starch-based multicomponent products (i.e., aquafeed) resulted in inferior physical quality of the extrudates [18,19]. In particular, the sectional expansion decreased and longitudinal expansion increased due to the addition of RPC.

When fruit press cakes were added to a starch matrix, lower sectional expansion of extrudates was found in previous studies [20,21,22,23]. A sharp decrease in sectional expansion was also reported by Wang and Kowalski [23] when 15 g/100 g cherry pomace (a by-product of cherry juice processing) was added to a corn starch matrix in twin-screw extrusion and by Day and Swanson [24] when whey protein, gluten or soy protein were added to extruded starch-based products. Accompanied by a decreased sectional expansion, some studies report a decreased hardness when the protein content was increased in starch-based extrudates [25,26]. Decreased sectional expansion and hardness of extrudates was attributed to change in rheological properties in the starch melt due to the addition of proteins or other particles (i.e., fibres) in the added component [27,28]. Moreover, in these studies, a decrease in sectional expansion due to the addition of high-protein components was accompanied by an increase in longitudinal expansion and was attributed to a decrease in viscous properties of the melt [23]. 

It can be assumed that with the addition of a high-protein and high-fibre ingredient such as RPC to starch, the rheological properties of the melt are highly influenced due to the reaction behaviour of RPC components at extrusion temperatures. When the rheological properties change due to RPC addition to starch, an alteration of the expansion mechanism (nucleation, bubble growth and bubble shrinkage) is expected compared to starch expansion. Additionally, the starch type might have an impact on rheological properties and therefore on expansion due to starch type-related physicochemical and functional properties. Rheological material properties (i.e., viscous and elastic properties) influence the time of water vapor pressure underrun in the die and determine the direction of expansion (sectional or longitudinal). 

Expansion in turn governs physical product characteristics like specific hardness, wherefore it can be assumed that with different expansion mechanisms, the physical product properties of starch or starch/RPC blends are affected by RPC inclusion. Furthermore, it can be assumed that the technofunctional product properties of starch and starch/RPC blends like water absorption and solubility depend on the addition of RPC and extrusion conditions and indicate underlaying physicochemical reactions of starch and RPC during extrusion. To enable the design of food products including RPC and starch, a systematic evaluation of the impact of RPC on expansion, rheological and technofunctional properties is required. 

In this study, we have evaluated the impact of the RPC content, starch type and extrusion temperature on the expansion, physical quality and extruder response of extruded starch/RPC blends. In order to draw a correlation between expansion mechanisms and rheological properties, the effect of RPC addition on the reaction behaviour and the complex viscosity of starch/RPC blends was investigated at extrusion-like conditions in a novel closed cavity rheometer. With the investigation of technofunctional properties of starch and starch/RPC blends, indications regarding physicochemical changes in starch and RPC during extrusion processing were collected. A list of abbreviations is provided in Table 1. 

## 2. Materials and Methods

### 2.1. Materials and Composite Flour Preparation

Cold-pressed 00 type RPC, rapeseed peel (RP) and rapeseed oil (RO) were kindly provided by Teutoburger Ölmühle (Ibbenbüren, Germany). The rapeseed used for RPC production was dehulled before deoiling. For all experiments, RPC and RP were milled to a particle size of < 0.5 mm as described by Martin and Osen [18]. Potato starch (PS) and waxy potato starch (WPS Eliane^TM^ 100) were kindly provided by Avebe (Veendam, The Netherlands). The amylose/amylopectin ratio of the starches were 25/75 for PS and 1/99 for WPS. 

PS and WPS were mixed with RPC, RP and RO in proportions of 70/10/10/10, 50/40/5/5 and 30/70/0/0 (*w/w*), respectively, to achieve rapeseed protein contents of 5, 16 and 27 g/100 g in the final product. RP and RO were added to the mixtures containing 50 and 70 g/100 g starch to achieve constant fibre and lipid contents in all the mixtures. 

The dry ingredients were mixed in a Spiral-Mixer SP 12 (DIOSNA Dierks & Söhne GmbH, Osnabrück, Germany) for 60 min. The mixtures were incubated at 20 °C for at least 8 h. Before extrusion, the dry matter in the mixtures was analysed in duplicate as laid down by legislation [29]. 

### 2.2. Extrusion 

The starch/RPC mixtures were extruded as laid down in [18] with the temperature settings reported in Table 2. The moisture content was set to 24 g/100 g or 29 g/100 g (dry matter basis), the mass flow rate was kept constant at 10 kg/h, the screw speed was set to 300 rpm and a 2 mm × 4.5 mm circular die was used. The screw was configured according to Tyapkova and Osen [19]. After extrusion, the extrudates were dried in an oven (Thermo Scientific Heraeus UT 6760, Thermo Electron LED GmbH, Langenselbold, Germany) at 40 °C for 24 h. The samples were stored in vacuum-sealed bags at 20 °C until further analyses. 

### 2.3. Determination of Extruder Response

As soon as steady-state operating conditions were reached, samples were taken. 

The average die pressure (bar) and product temperature in front of the die (°C) during the sample-taking period were recorded using CSpro medium software (Coperion GmbH, Stuttgart, Germany). 

The specific mechanical energy input (SME) was calculated as described in a number of recent studies [30,31].

### 2.4. Characterisation of the Raw Material and Extrudates

#### 2.4.1. Chemical Composition 

The chemical compositions of the starches, RPC, RP, starch/RPC mixtures and all extruded samples were analysed. Regarding the latter, the extrudates were ground after drying to < 0.5 mm particle size using a knife mill (Grindomix GM 200, Retsch GmbH, Haan, Germany). The dry matter (DM) was measured as laid down in the German Food Act [29]. The protein content was determined using the Dumas method as laid down in the German Food Act [29] using a TruMac N Protein Analyzer (LECO Corporation, St. Joseph, MI, USA) with a conversion factor of 6.25. The lipid content was determined in accordance with Caviezel, DGF K-I 2c (00) [32]. The ash content (at 950 °C) was determined thermogravimetrically in accordance with AOAC International [33]. The crude fibre content was analysed with a Fibretherm (Gerhardt GmbH & Co. KG, Königswinter, Germany) in accordance with AOAC International [34]. Soluble and insoluble fibre contents of RPC were analysed by enzymatic-gravimetric analysis according to AOAC 991.43. The starch content was determined using the hexokinase method [35]. All analyses were at least carried out in duplicate. 

#### 2.4.2. Particle Size Distribution

The particle size distribution of the RPC, RP and starches was determined using a Malvern laser diffraction particle size analyser (Mastersizer S Long Bed Version 2.15, Malvern Instruments Ltd, Malvern, UK) as described by Osen and Toelstede [36]. Analyses were carried out in duplicate.

### 2.5. Physical Properties of Extrudates

#### 2.5.1. Expansion 

Sectional and longitudinal expansion indices (SEI and LEI) were calculated according to Alvarez-Martinez and Kondury [37] and Carvalho and Takeiti [38]. The volume expansion index (VEI) is the product of LEI and SEI. 

#### 2.5.2. Specific Hardness

For texture analyses, 10 randomly chosen extrudate strands were measured with a digital calliper to determine their length l_E_ and diameter d_E_ before being placed on the measuring platform of a texture analyser (TA.XTplus, Stable Micro Systems, Surrey, UK) equipped with a 50 kg measuring cell. A cylindrical punch (SMS P/25L) compressed the samples to 30% deformation at a test speed of 8 mm s^−1^ until breakage. The applied force (F_max_) on the lateral surface of the extrudate strands is the specific hardness (N/mm^2^).
H_spec_ [N⁄mm^2^] = F_max_/(d_E_ × l_E_ × π).(1)

### 2.6. Water Absorption and Water Solubility 

The water absorption was determined following the method of 56-20.01 with slight modification [39]. For this, 2 g of powder (m_Sample_) was placed in a 50 mL centrifuge tube (m_Centr_) and mixed with 40 mL demineralised water. After 10 min stirring with a reagent shaker (VF 2, Janke & Kunkel, IKA-Labortechnik GmbH, Staufen, Germany), the samples were centrifuged for 10 min at 1000 relative centrifugal force (RCF) and 20 °C using a 6K15 centrifuge (Sigma Laborzentrifugen GmbH, Osterode am Harz, Germany). The supernatant was decanted into a tared aluminium dish (m_Alu_), dried in an oven at 105 °C for 18 h (Heraeus UT 6060, Thermo Electron LED GmbH, Langenselbold, Germany) and then weighed after drying (m_Alu + Solids_). The residual sediment in the centrifuge tube was also weighed (m_Centr + Sed_). The water absorption index (WAI) was calculated using Equation (2).
WAI [g/g]= (m_(Centr + Sed)_ − m_Centr_)/m_sample_(2)

The water solubility index (WSI) was calculated according to Equation (3) and described the amount of dry solids in the supernatant relative to the original sample mass [40]. The samples were analysed in duplicate.
WSI [%]= (m_(Alu + Solids)_ − m_Alu_)/m_Sample_ × 100(3)

### 2.7. Rheological Properties 

The impact of the thermal treatment on structure formation in the starches, RPC and starch/RPC mixtures were investigated using a closed cavity rheometer (RPA flex, TA Instruments, New Castle, Delaware, USA) as described by Quevedo and Jandt [41]. Before the rheological analyses, the dry powders were mixed with demineralised water to obtain moisture contents of 24 or 29 g/100 g (corresponding to the moisture conditions in the extruder). After mixing, the samples were stored at 4 °C for at least 8 h to allow adequate moisture distribution and brought to room temperature before analysis. For the oscillatory tests, 6 g of sample were placed between two grooved and thermoregulated cones. The lower cone oscillated and a transducer in the upper cone received the torque response of the sample, allowing rheological properties such as the complex modulus |G*|, storage modulus G’, loss modulus G’’ and complex viscosity |ƞ*| to be calculated. The complex modulus |G*| describes the stress–strain relationship of a linear viscoelastic material. The ratio of |G*| to the frequency gives the complex viscosity of a sample. Experiments were conducted in duplicate and on the LVE region (data not shown). 

The impact of thermal treatment on the reaction onset temperature was evaluated by temperature sweep analyses in a range from 30 to 140 °C (heating rate 5 K/min, $\dot{\gamma }$ = 0.1 s^−1^). 

To evaluate the impact of extrusion on the rheological properties of the samples, extruded samples were treated 10 s at a shear rate of $\dot{\gamma }$ = 31 s^−1^, before they were kept at a constant low shear rate of $\dot{\gamma }$ = 0.1 s^−1^ for 8 min. The pretreatment and measurement temperatures T_Pre_ and T_M_ were 100, 120 or 140 °C. 

### 2.8. Statistical Analysis 

All experiments were at least performed in duplicate and were given as mean values with standard errors. The effects of starch/RPC ratio and process conditions were determined by one-way analysis of variance (ANOVA) with a significance threshold of *p* < 0.05. Where appropriate, mean values were compared using Tukey’s honest significance test using Origin Lab Software (OriginLab, Northampton, MA, USA).

## 3. Results

### 3.1. Chemical Composition

The composition of the RPC, RP, PS, WPS and RPC/starch mixtures is summarised in Table 3. Table 4 reports the dry matter content (g/100 g) of the extruded and dried samples. 

The data show that RPC had a protein content of 38.2 g/100 g. Previous studies reported lower RPC protein contents (18.6–30.0 g/100 g) [2,5,7]. The comparatively high protein content of the RPC used in this study might be due to the protein-rich *Brassica napus* L. variety our supplier processes. Furthermore, peeling the rapeseed before deoiling results in a comparatively low fibre content, increasing the RPC protein content. However, the protein content of the RPC that was used for this work lies within the range of that of rapeseed meals previously studied (34.4–40.6 g/100 g) [4]. 

The RP obtained from the dehulling of the rapeseed had a lower protein content and higher lipid content than RPC (15.7 g/100 g protein and 25.5 g/100 g lipid). The fibre content of RPC (4.7 g/100 g) was much lower than the fibre content of the RP (29.4 g/100 g). 

In the RPC/starch mixtures, the protein content increases from 5.4 to 26.7 g/100 g with increasing RPC content, whereas the starch content decreases from 69.0 to 29.6 g/100 g. The lipid and fibre contents of the RPC/starch mixtures were kept constant by adding small amounts of RO and RP. PS and WPS had the smaller particle sizes than RPC and RP. 

### 3.2. Physical Properties of the Extrudates 

#### 3.2.1. Expansion

As expansion phenomena is linked to flash evaporation at > 100 °C barrel temperature, the expansion indices of samples extruded at such temperatures are discussed here. 

Figure 1a shows that the addition of RPC to potato starch at 40 g/100 g increased the SEI, but decreased the LEI and VEI, significantly. A further increase in the RPC content to 70 g/100 g resulted in an even higher SEI, whereas LEI was not further influenced. This resulted in a product with higher VEI compared to the sample without RPC. The results with WPS depicted in Figure 1b show a different response to the RPC addition. Here, the addition of 40 g/100 g RPC leads to lower SEI and VEI but higher LEI compared to samples with only 10 g/100 g RPC. Samples containing 70 g/100 g RPC exhibited a larger SEI and VEI than samples containing 40 g/100 g RPC, whereas the LEI only slightly increased. 

The increase in temperature has mainly led to an increase in SEI, LEI and VEI in PS samples containing 70 g/100 g RPC. LEI and VEI were lowest for 40 g/100 g RPC samples extruded at 100 °C. In WPS samples, a barrel temperature of 100 °C resulted in higher SEI and VEI regardless of RPC content, but did not have a large impact on LEI. 

It is expected that the addition of protein- and fibre-rich biopolymers such as press cakes to starch influences the rheological properties of the melt [42], which further affects the expansion behaviour of the matrix. A number of experimental studies describe the close correlation of the rheological properties of biopolymers and the corresponding expansion behaviour in extrusion [42,43,44,45,46]. It was found that the viscosity of the melt is associated with axial expansion and LEI, whereas melt elasticity is correlated with radial expansion and therefore the SEI [37]. In our study, we did not differentiate between viscous and elastic properties in rheological analyses, but showed that the starch/RPC blends exhibited lower |G*| with increasing RPC content and temperature. An alteration of rheological properties due to RPC addition associated with an increase in LEI and a decrease in SEI was, i.e., shown in WPS-based samples with 40 or 70 g/100 g RPC, respectively. 30WPS/70RPC showed the lowest |G*| in rheological analyses (discussed later, Figure 4) and the highest LEI at 29 g/100 g moisture content.

The observed decrease in SEI due to 40 g/100 g RPC addition in WPS samples in our study is in accordance to previous studies that found lower sectional expansion of extrudates when fruit press cakes or other by-products were added to a starch matrix [20,21,22,23]. The authors of studies on bilberry, apple or blackcurrant press cake attributed this to the presence of dietary fibre in the press cakes. Especially the presence of inert insoluble fibre particles was associated with a strong physicochemical incompatibility with starch, a decrease in viscosity and a reduced SME that overall resulted in a decreased SEI [22,23,47,48]. Furthermore, insoluble fibre particles tend to rupture the cell walls before the gas bubbles extend to their full size [49].

The RPC used in this study contains 35.67 ± 5.19 g/100 g total dietary fibre, whereof 88.3% are insoluble and 11.7% are soluble. The impact of extrusion on the dietary fibre content of RPC will be part of future studies. Since with RPC addition, the amount of insoluble fibre in the blends, which act mainly as filler particles, increases markedly, the decrease in SEI in sample WPS40/RPC50 might be explained. However, the high SEI and VEI of samples with 70 g/100 g do not agree with this hypothesis. It can be assumed that high SEI and VEI in RPC-rich samples are driven by swelling expansion mechanisms more than by sudden water evaporation and may be associated with the presence of soluble fibres and an overall high water binding and holding capacity of RPC compared to starch. Soluble fibre fractions have been shown to exhibit large expansion volumes due to their greater water-binding capacity, which enables more nucleation sites for water vapor to develop at die exit [48,50]. Many authors have described that the high shear force in the extruder barrel might mechanically break parts of the insoluble fibre fractions, which increases the fraction of soluble dietary fibre and increases the water binding of extruded materials compared to unextruded materials [51]. We assume that parts of the insoluble fibre fractions in RPC are solubilized during extrusion processing and enhance swelling expansion, accompanied by an increase in WAI of extruded samples. Results in Section 3.2.3 confirm that RPC-rich samples show increased WAI due to extrusion treatment. 

Unfortunately, the extrusion of 70PS/10RPC resulted in severe extruder clogging, wherefore no coherent extrudate strains were generated and the direct measurement of expansion indices was not possible. This means that the impact of starch type on the expansion of samples containing 10 g/100 g RPC cannot be compared here. However, the increase in RPC from 40 to 70 g/100 g increased the SEI and VEI of both PS and WPS based samples. 

At 140 °C barrel temperature, PS samples containing 70 g/100 g RPC showed a higher SEI compared to WPS-based samples. This effect cannot be explained with our results, but might be due to a low gelatinisation temperature, large granule size and favourable amylose content of PS compared to WPS that allowed bubble growth and extrudate fixation and enhanced the sectional expansion [52]. 

#### 3.2.2. Specific Hardness 

The specific hardness of the samples is shown in Figure 2. For WPS samples, the specific hardness of the extrudates decreased with increasing RPC content. Similar results were observed with PS samples regardless of the barrel temperature and moisture content. In WPS as single component, a higher moisture content significantly increased the hardness, but in samples containing RPC, this effect was not observed. 

Furthermore, although extruded PS was softer than WPS, no difference in specific hardness was seen between the two starch types when RPC was added, which was also seen for the correlation between starch type and expansion. The hardest extrudates were found for PS and WPS extruded at a moisture content of 29 g/100 g and 120 or 100 °C barrel temperature, respectively. 

RPC contains a number of components that might have a large impact on the specific hardness of the extrudates, such as proteins, fibres, sugars and lipids. 

The results suggest that one reason for a decrease in hardness with increasing RPC content in our study might be the relatively high protein content of RPC. The work of Zhu and Shukri [26] showed that the addition of 20 g/100 g soy protein concentrate to corn starch extrudates resulted in a decreased hardness of the product, which is in accordance to our results. The same trend was found by [25], where 25 g/100 g whey protein in starch-based extrudates resulted in a decrease in hardness. However, a decrease in hardness was often accompanied by a decreased SEI in these studies. In our work, RPC-rich samples exhibited a high SEI and VEI but low specific hardness. Therefore, we assume that the swelling expansion mechanisms described before generated a porous structure that did not require large breaking forces to be mechanically disrupted. Furthermore, the soluble fibre fractions in RPC, mostly pectins, hemicelluloses, mixed β-glucans, gums and mucilages [53,54], might have softened the structure of the expanded RPC-rich extrudates, especially under the assumption that parts of the insoluble fibre fractions have been solubilised due to extrusion processing. This in in alignment with previous studies who reported that pectin and other soluble fibres have been shown to decrease specific hardness of directly expanded snack products [55]. 

#### 3.2.3. Water Absorption Index and Water Solubility Index

Table 5 summarises the WAI of PS and WPS mixtures for different extrusion moisture contents and temperatures > 100 °C. In both PS/RPC and WPS/RPC mixtures, the WAI increased with increasing barrel temperature. The opposite was found for PS and WPS as single ingredients. Untreated mixtures with 70 g/100 g RPC had higher WAIs than starch-rich mixtures. 

In starch-rich samples, the temperature-dependent WAI increase was greater than in RPC-rich samples. In PS and WPS samples extruded at 140 °C, the WAI decreased with increasing RPC content, whereas extruded PS and WPS as single materials exhibited low WAI.

Water absorption is dependent on the availability of hydrophilic groups that are able to bind water, on capillary forces and on the porosity of the matrix [27]. 

The high WAI of unextruded RPC-rich samples may be attributed to the water binding ability of fibre components in the RPC, as has been investigated in earlier studies [56]. Furthermore, the increase in WAI due to increased barrel temperature in samples containing RPC can be attributed to a modification of the morphology (i.e., particle size and shape) of fibres present in RPC due to heat and shear treatment during extrusion. Arrigoni and Caprez [57] and Zhang and Bai [58] found that the swelling capacity of extruded depectinised apple pomace or oat bran was increased after extrusion. The decrease in WAI in extruded starch can be attributed to heat and shear during extrusion. Heat in the extruder barrel causes the starch to lose its crystalline structure and shear leads to molecular fragmentation, both factors leading to a homogenous “melt” and a decrease in the WAI. 

The WSI of the PS and WPS mixtures increased with increasing RPC content at all barrel temperatures and at both moisture contents (Table 6). The WSI of PS/RPC and WPS/RPC mixtures decreased as the barrel temperature was increased from 20 to 140 °C, hence this effect was stronger for RPC-rich samples compared to starch-rich mixtures.

Although PS and WPS were not soluble at room temperature (WSI of PS = 0.5 and WSI of WPS = 0.75), the extrusion process markedly increased the WSI of the samples.

The starch type and moisture content during the extrusion process did not have a large impact on the WSI of samples. 

For starch-based matrices, the water solubility (WSI) is often correlated to the extent of degradation of the molecular components of the starch. In multicomponent systems, the WSI indicates the amount of components (polysaccharides, proteins and fibres) that can be solubilised after mechanical breakdown resulting from, e.g., high shear forces in the extruder [59,60]. 

A higher RPC content in the formulations leads to an increase in rapeseed protein from 5 g/100 g in the 70PS/10RPC and 70WPS/10RPC mixtures to 26.7 g/100 g in the 30PS/70RPC and 30WPS/70RPC mixtures. The main protein fractions of rapeseed—cruciferin and napin—are soluble at room temperature at neutral pH [61,62]. This explains the higher WSI of RPC-rich samples compared to starch-rich samples. An increased WSI with increasing protein content of starch/protein formulations has also been reported by Zhu and Shukri [26]. The marked WSI increase in PS and WPS as single ingredients may be attributed to a greater extent of macromolecular degradation due to higher barrel and melt temperatures, respectively, as described in previous research [63]. In general, the WAI and WSI correlated negatively at high barrel temperatures in this study. This trend agrees with other literature [27,64,65].

### 3.3. Rheological Properties 

#### 3.3.1. Reaction Behaviour of RPC and RPC/Starch Blends before Extrusion 

The complex modulus |G*| of the starches, RPC and their mixtures were measured as samples were heated in the CCR rheometer (Figure 3 and Figure 4). The values of |G*| indicate that structural changes occur in the biopolymers as a function of temperature [41]. Of all the samples investigated, RPC had the lowest |G*| and the value decreased as the temperature increased from 30 to 50 °C. Upwards of 50 °C, there was a slight increase in |G*| followed by a steep increase as the temperature reached 100 °C (Figure 3). In contrast, the PS/RPC and WPS/RPC mixtures showed no increase in |G*| in that temperature region (Figure 4a,b). It can be assumed that the slight increase in |G*| for the RPC starting at 70 °C corresponds to the onset of rapeseed protein denaturation. The most abundant protein types in rapeseed (*Brassica napus* canola) are the 11S globulin cruciferin and the 1.7–2S albumin napin, representing, respectively, 60% and 20% of the total accumulated protein [66]. It has been reported that at temperatures of 65–70 °C, the hexameric quaternary structure of cruciferin unfolds, weakening the matrix structure as indicated by a decrease in |G*| [66]. Due to the resulting increase in the number of exposed reactive binding sites, new linkages occur that lead to larger molecular size or mass. However, these structure-forming processes (i.e., aggregation) lead to higher viscosity and increase |G*| accordingly, as seen by the small increase above 70 °C and large increase above 100 °C [41,67,68,69]. Previous studies have reported thermal transition peaks at 91 and 110 °C for purified cruciferin and napin, respectively. Studies on rapeseed protein isolate found reaction onset at 84 °C for cruciferin and 102 °C for napin [11]. Although the purification method, protein concentration and the method of thermal analysis has a large impact on the measured thermal transition, these findings indicate that the presence of nonprotein and other protein components affect the thermal stability of rapeseed materials [70]. This is in line with our results, as RPC (comparable to rapeseed flour) contains a number of nonprotein components (i.e., fibre and lipid) that may reduce the thermal stability of rapeseed proteins and therefore accelerate the reaction onset temperature. 

Between 100 and 120 °C, the complex modulus |G*| of RPC increases indicating structure formation induced by aggregation reactions. At temperatures above 120 °C, the complex modulus |G*| decreases. There are two possible explanations for this. The aggregation reactions might be complete and |G*| may decrease as a consequence of higher molecular mobility [41]. Alternatively, the newly formed structure of RPC no longer remains intact with disintegration and deaggregation reactions destabilising the matrix [11]. We assume that |G*| decreases as a consequence of increased temperature, as indicated by an increasing |ƞ*| in RPC-rich extruded samples (see Section 3.3.2). 

The PS and WPS as single ingredients both showed a steep and constant decrease in |G*| as the temperature increased from 30 to 70 °C. At > 70 °C, |G*| decreased less steeply and at > 105 °C abruptly. In contrast to WPS with constant |G*| in the 120–140 °C region, PS showed a marked increase in |G*| starting at 120 °C. Overall, the complex modulus throughout temperatures between 30 and 110 °C was higher for PS compared to WPS. This can be attributed to the higher amylose content of PS (25 g/100 g amylose) compared to WPS (1 g/100 g amylose). A high amylose content can lead to increased entanglements between the linear amylose chains when starchy melts are heated resulting in a higher viscosity [71].

#### 3.3.2. Complex Viscosity of Extruded RPC/Starch Blends 

Figure 5 illustrates the complex viscosity |ƞ*| of extruded PS/RPC blends as a function of time and temperature. At 100 °C, T_Pre_ and T_M_, PS/RPC70 is extruded at 100 °C exhibited the highest |ƞ*|, followed by PS/RPC70 extruded at 140 °C and PS/RPC40 extruded at 120 °C. PS extruded at 120 °C exhibited the lowest |ƞ*|. The |ƞ*| of PS/RPC70 extruded at 100 °C increased over time, when T_Pre_ and T_M_ was 100 °C, which indicates that aggregation reactions in RPC that were induced by extrusion are not completed. PS/RPC70 extruded at 140 °C only shows a slight curve slope, wherefore we assume that a higher T_B_ during extrusion results in a higher degree of aggregation reactions. The comparatively low |ƞ*| of PS extruded at 120 °C and the small impact an increase in T_Pre_ and T_M_ (100 or 120 °C) indicates that the starch was gelatinized during extrusion and constitutes an amorphous melt during rheological analysis. 

To mimic the rheological properties of the melt at the moment of die exit, T_Pre_ and T_M_ were set to the corresponding T_B_ to which the samples were applied to during extrusion. |ƞ*| decreased in the order PS/RPC70 extruded at 140 °C, PS/RPC40 extruded at 120 °C and PS extruded at 120 °C. A high viscosity during extrusion can be associated with a larger SEI, since high viscosity results in high pressure at the die, wherefore the vapour pressure is undergoing later during expansion and the matrix can facilitate its structure during bubble growth. A low viscosity can in contrast be associated with a more pronounced LEI. In our study, for PS samples, the SEI increases and LEI decreased with an increasing RPC content. These observations can be linked to |ƞ*| in rheological analyses and indicate that a closed cavity rheometer can be used as a tool to find indications regarding the expansion properties of biopolymers. 

### 3.4. Extruder Response

The extruder response is shown in Table 7 and Table 8. At a barrel temperature of 20 °C, starch-rich samples resulted in ca. 10 °C higher product temperatures (62.7 °C) than RPC-rich mixtures (52.2–55.0 °C). Hence, when the barrel temperature was set to 80, 100 and 120 °C, mixtures with 70 g/100 g RPC generated the highest product temperature of all the mixtures. When extruded at 140 °C, no formulation-related difference in product temperature was observed. Almost no difference in SME was observed between PS or WPS based samples having different moisture and RPC contents. In mixtures containing WPS, the pressure decreased with increasing barrel temperature regardless of the RPC content, whereas WPS-rich samples generated a higher pressure at the die (40.51 bar at a moisture content of 24 g/100 g) than RPC-rich samples (12.90 bar at a moisture content of 24 g/100 g). The lowest pressure was found for the 50WPS/40RPC sample.

Starch might cause higher viscous dissipation and therefore increase the product temperature due to its high shear viscosity at low temperatures as described, i.e., by Ye and Hu [72]. 

The pressure at the die is a result of the melt viscosity and is, in turn, influenced by shear and the thermal input in the extruder. Rheological investigations showed, e.g., that the 70WPS/10RPC sample had a dynamic viscosity of 179.5 kPas when treated at 80 °C for 1 min, whereas the 30WPS/70RPC sample had a dynamic viscosity of 54.5 kPas (data not shown). The higher viscosity of the extruded 70WPS/10RPC melt leads to higher die pressure.

## 4. Conclusions

The aim of this study was to investigate the impact of rapeseed press cake (RPC) concentration on expansion, specific hardness, water binding and solubility, rheological properties and extruder response using a suitable model system. To correlate expansion behaviour with the rheological properties of the melt, a specific rheometer simulating extrusion conditions was used to evaluate the rheological properties of the extruded blends. 

Starch was chosen as the basis ingredient and the press cake content was varied far beyond those concentrations used in previous studies. 

Results gained in a specific rheometer show that the addition of RPC had a large effect on the complex viscosity of starch/RPC blends when treated at extrusion like conditions. The RPC increased the complex viscosity of the starch/RPC blends, which could be linked to a more pronounced sectional and decreased longitudinal expansion of the samples. 

It was possible to extrude starch/RPC mixtures containing up to 70 g/100 g RPC at barrel temperatures of 100–140 °C and moisture contents of 24 or 29 g/100 g. The physical properties of the extrudates were highly dependent on barrel temperature and RPC content, whereas starch type and moisture content did not have a large impact on expansion or texture in the range investigated. At temperatures above 120 °C, 70 g/100 g of RPC increased the sectional and volumetric expansion of extrudates, irrespective of starch type. At the same temperatures and RPC inclusion level, the longitudinal expansion decreased when RPC was added to potato starch and slightly increased when blends contained RPC and waxy potato starch. Moreover, 40 and 70 g/100 g RPC decreased the hardness and water absorption of the extrudates and increased the water solubility. Especially the extrusion of native potato starch/RPC at a high RPC content and high barrel temperatures resulted in a high expansion.

In this study, we were able to correlate expansion behaviour and rheological properties using a closed cavity rheometer. However, expansion properties were only physically investigated > 10 s after die exit, whereas it is known from previous studies, that expansion comprises several stages of initial growth and shrinkage. To give a more detailed causal link between material properties and expansion behaviour during all expansion phases, digital imaging at the die exit should be used and the rheological investigations should be extended to the viscous and elastic modulus of the blends under thermomechanical treatment. 

## Figures and Tables

**Figure 1 polymers-13-00215-f001:**
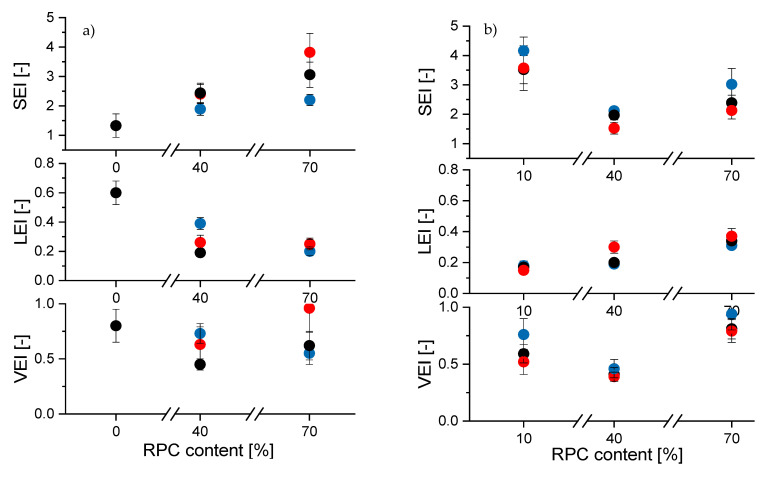
Sectional, longitudinal and volumetric expansion indices (SEI, LEI, and VEI) of (**a**) potato starch (PS) and (**b**) waxy potato starch (WPS) and their mixtures with 10, 40 and 70 g/100 g rapeseed press cake (RPC) extruded at a moisture content of 29 g/100 g and barrel temperatures of 100 °C (
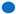
), 120 °C (
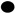
) and 140 °C (
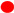
). Values are shown as mean values with standard deviations based on 10 duplicates.

**Figure 2 polymers-13-00215-f002:**
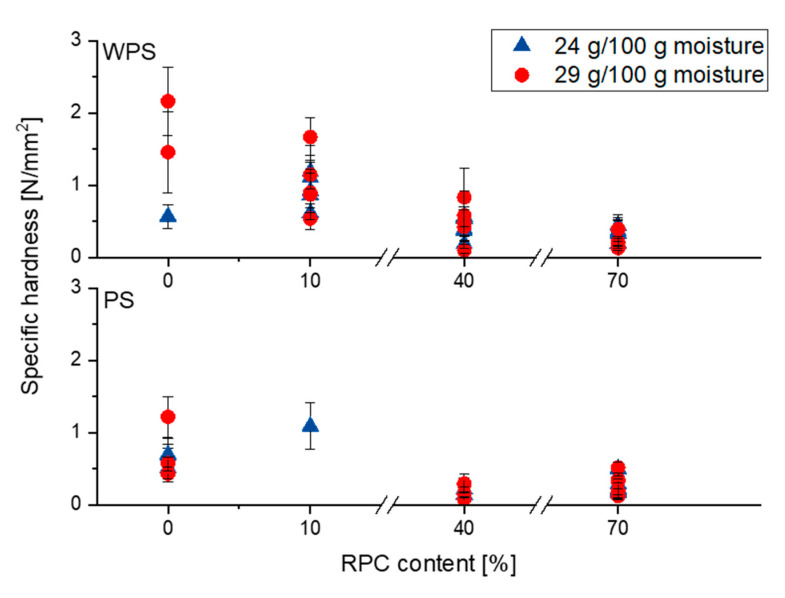
Specific hardness of waxy potato starch (WPS) and potato starch (PS) and their mixtures with 0, 10, 40 and 70 g/100 g rapeseed press cake (RPC) extruded at 24 (
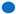
) and 29 (
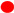
) g/100 g moisture content and barrel temperatures of 100, 120 and 140 °C. Values are shown as mean values with standard deviations based on 10 duplicates.

**Figure 3 polymers-13-00215-f003:**
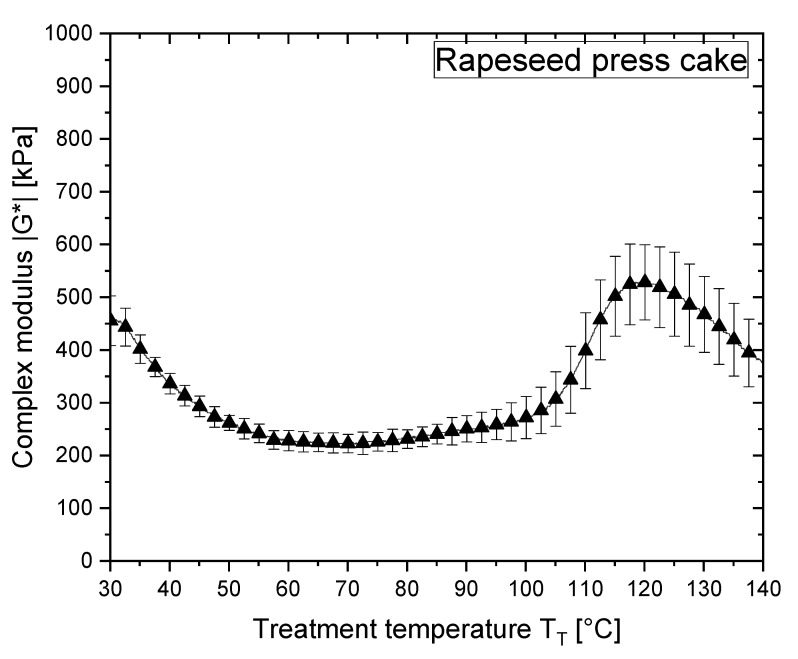
Complex modulus |G*| as a function of temperature (30–140 °C; heating rate 5 K/min) at a shear rate of $\dot{\gamma }$ = 0.1 s^−1^ for rapeseed press cake (RPC) at a moisture content of 24 g/100 g.

**Figure 4 polymers-13-00215-f004:**
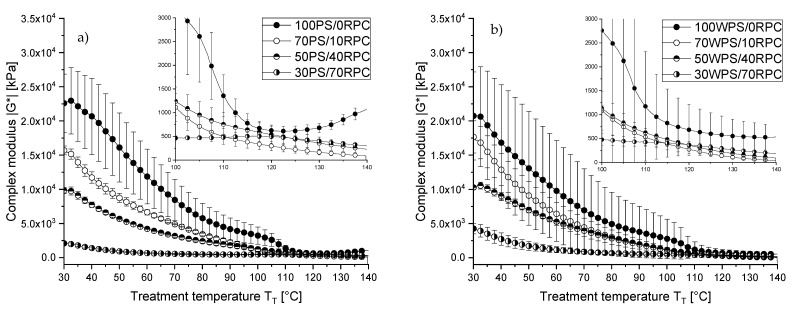
Complex modulus |G*| as a function of temperature (30–140 °C; heating rate 5 K/min) at a shear rate of $\dot{\gamma }$ = 0.1 s^−1^ for (**a**) potato starch (PS) and (**b**) (WPS) mixed with rapeseed press cake (RPC) at a moisture content of 24 g/100 g.

**Figure 5 polymers-13-00215-f005:**
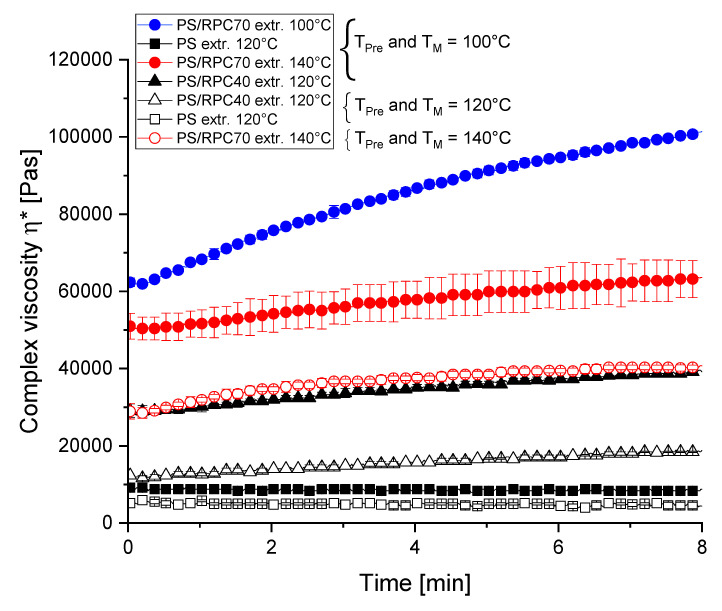
Complex viscosity |ƞ*| as a function of time and temperature (100 °C (
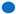
), 120 °C (
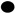
) and 140 °C (
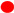
)) of potato starch (PS) singularly and in blends with 40 or 70 g/100 g rapeseed press cake (RPC). Samples extruded and analysed at 29 g/100 g moisture content. Pretreatment 10 s at $\dot{\gamma }$ = 31 s^−1^; measurement 8 min at $\dot{\gamma }$ = 0.1 s^−1^.

**Table 1 polymers-13-00215-t001:** List of abbreviations.

DM	Dry matter (g/100 g)
H_spec_	Specific hardness
LEI	Longitudinal expansion index
n.a.	Not analysed
NFE	Nitrogen free extract (g/100 g)
PS	Potato starch
RO	Rapeseed oil
RP	Rapeseed peel
RPC	Rapeseed press cake
SEI	Sectional expansion index
SME	Specific mechanical energy (Wh/kg)
T	Temperature (°C)
T_B_	Barrel temperature (°C)
T_M_	Temperature of measurement (°C)
T_Pre_	Temperature of Pre-treatment (°C)
T_T_	Treatment temperature (°C)
VEI	Volumetric expansion index
WAI	Water absorption index
WPS	Waxy potato starch
WSI	Water solubility index

**Table 2 polymers-13-00215-t002:** Extrusion barrel temperature settings for extruded starches and starch/rapeseed press cake (RPC) mixtures.

	Temperature (°C)					
Barrel Segment	1	2	3	4	5	6
**Sample 20 °C**	–	20	20	20	20	20
**Sample 80 °C**	–	60	80	80	80	80
**Sample 100 °C**	–	60	80	100	100	100
**Sample 120 °C**	–	60	80	100	120	120
**Sample 140 °C**	–	60	100	120	140	140

**Table 3 polymers-13-00215-t003:** Chemical composition and particle size of rapeseed press cake (RPC), rapeseed peel (RP), potato starch (PS), waxy potato starch (WPS) and the mixtures of starch with 10, 40 or 70 g/100 g rapeseed press cake (RPC).

Raw Material (g/100 g)	DM (g/100 g)	Protein × 6.25 (%DM)	Lipid (%DM)	Raw Fibre (%DM)	Ash (%DM)	Starch (%DM)	Particle Size d_50.3_ (µm)
**RPC**	95.1± 0.03	38.2 ± 0.30	23.4 ± 0.90	4.7 ± 0.03	7.3 ± 0.02	3.00 ± 0.02	261.1 ± 4.5
**RP**	93.6 ± 0.18	15.7 ± 0.08	25.5 ± 1.12	29.4 ± 0.31	4.1 ± 0.03	4.39 ± 0.03	418.9 ± 15.9
**PS**	81.9 ± 0.02	n.a.	n.a.	n.a.	n.a.	97.6 ± 0.40	40.3 ± 1.2
**WPS**	80.5 ± 0.07	n.a.	n.a.	n.a.	n.a.	91.7 ± 0.25	38.1 ± 0.9
**Calculated chemical composition (g/100 g)**							
**30PS/70RPC**	91.1	26.7	16.4	3.3	n.a.	32.4	n.a.
**50PS/40RPC**	88.7	16.0	15.6	3.4	n.a.	50.2	n.a.
**70PS/10RPC**	86.2	5.4	14.9	3.4	n.a.	69.0	n.a.
**30WPS/70RPC**	90.7	26.7	16.4	3.3	n.a.	29.6	n.a.
**50WPS/40RPC**	88.0	16.0	15.6	3.4	n.a.	47.3	n.a.
**70WPS/10RPC**	85.2	5.4	14.9	3.4	n.a.	64.9	n.a.

**Table 4 polymers-13-00215-t004:** Dry matter content of potato starch (PS) and waxy potato starch (WPS) mixtures with 10, 40 or 70 g/100 g rapeseed press cake (RPC) after extrusion processing at 24 or 29 g/100 g moisture content and a barrel temperature of 20, 80, 100, 120 or 140 °C and drying (40 °C, 24 h).

	Barrel Temperature (°C)					
24 g/100 g moisture	Not extruded	20	80	100	120	140
**PS**	90.4 ± 0.02	90.5 ± 0.04	92.1 ± 0.12	87.9 ± 0.11	88.6 ± 0.03	86.2 ± 0.04
**70PS/10RPC**	91.2 ± 0.01	91.5 ± 0.02	92.3 ± 0.02	88.3 ± 0.04	89.8 ± 0.12	88.5 ± 0.05
**50PS/40RPC**	86.9 ± 0.02	92.5 ± 0.03	92.3 ± 0.07	92.5 ± 0.05	89.9 ± 0.04	91.3 ± 0.15
**30PS/70RPC**	91.9 ± 0.04	91.7 ± 0.11	95.5 ± 0.03	93.2 ± 0.06	91.9 ± 0.18	95.8 ± 0.14
**WPS**	92.5 ± 0.03	92.1 ± 0.09	91.6 ± 0.04	90.7 ± 0.05	–	–
**70WPS/10RPC**	89.9 ± 0.03	90.8 ± 0.05	88.5 ± 0.05	88.6 ± 0.11	89.9 ± 0.15	90.1 ± 0.05
**50WPS/40RPC**	90.0 ± 0.02	92.3 ± 0.02	90.0 ± 0.02	88.6 ± 0.09	90.0 ± 0.04	89.8 ± 0.06
**30WPS/70RPC**	91.1 ± 0.02	92.2 ± 0.04	93.0 ± 013	91.8 ± 0.11	92.3 ± 0.11	91.2 ± 0.02
29 g/100 g moisture	Not extruded	20	80	100	120	140
**PS**	91.1 ± 0.07	90.6 ± 0.02	92.3 ± 0.06	88.7 ± 0.18	89.6 ± 0.12	90.1 ± 0.08
**70PS/10RPC**	92.0 ± 0.06	92.1 ± 0.09	92.8 ± 0.12	89.0 ± 0.08	90.4 ± 0.06	89.8 ± 0.09
**50PS/40RPC**	87.1 ± 0.08	93.2 ± 0.20	93.0 ± 0.07	93.0 ± 0.14	90.0 ± 0.05	91.3 ± 0.14
**30PS/70RPC**	92.1 ± 0.08	92.7 ± 0.01	97.8 ± 0.04	94.3 ± 0.01	92.6 ± 0.04	96.6 ± 0.11
**WPS**	92.5 ± 0.02	92.1 ± 0.15	91.6 ± 0.14	90.7 ± 0.12	–	–
**70WPS/10RPC**	89.9 ± 0.04	90.8 ± 0.03	88.5 ± 0.02	88.6 ± 0.01	89.9 ± 0.04	90.1 ± 0.12
**50WPS/40RPC**	90.0 ± 0.01	92.3 ± 0.09	90.0 ± 0.06	88.6 ± 0.06	90.0 ± 0.20	89.8 ± 0.08
**30WPS/70RPC**	91.1 ± 0.11	92.2 ± 0.15	93.0 ± 0.07	91.8 ± 0.06	92.3 ± 0.16	91.2 ± 0.13

**Table 5 polymers-13-00215-t005:** Water absorption index (WAI) of potato starch (PS) and waxy potato starch (WPS) and their mixtures with 10, 40 and 70 g/100 g rapeseed press cake (RPC) extruded at 24 and 29 g/100 g moisture content and barrel temperatures of 20, 80, 100, 120 and 140 °C. Mean values with different superscript letters (a, b, c, d) within one column (starch type and moisture content) indicate significant differences (*p* > 0.05) based on a one-way analysis of variance (ANOVA). Where appropriate, the mean values were compared using Tukey’s honest significance test.

Mixture	Untreated	Moisture Content (g/100 g)	Barrel Temperature				
			20 °C	80 °C	100 °C	120 °C	140 °C
PS	1.67 ± 0.01 ^a^	24	–	1.78 ± 0.25 ^a^	0.59 ± 0.04 ^a^	0.58 ± 0.17 ^a^	0.86 ± 0.13 ^a^
70PS/10RPC	1.78 ± 0.02 ^a^	24	2.84 ± 0.04 ^a^	2.69 ± 0.01 ^b^	4.63 ± 0.02 ^b^	4.63 ± 0.01 ^b^	8.16 ± 0.10 ^b^
50PS/40RPC	2.29 ± 0.01 ^b^	24	2.60 ± 0.01 ^b^	2.70 ± 0.02 ^b^	3.14 ± 0.02 ^c^	3.45 ± 0.02 ^c^	5.99 ± 0.10 ^c^
30PS/70RPC	2.88 ± 0.08 ^c^	24	3.11 ± 0.01 ^c^	3.17 ± 0.02 ^b^	4.17 ± 0.02 ^d^	4.83 ± 0.35 ^b^	5.53 ± 0.01 ^d^
PS	1.67 ± 0.01 ^a^	29	–	1.05 ± 0.27 ^a^	1.77 ± 0.04 ^a^	1.57 ± 0.08 ^a^	0.96 ± 0.24 ^a^
70PS/10RPC	1.78 ± 0.02 ^a^	29	2.78 ± 0.03 ^a^	3.71 ± 0.02 ^b^	4.41 ± 0.00 ^b^	5.05 ± 0.04 ^b^	7.73 ± 0.05 ^b^
50PS/40RPC	2.29 ± 0.01 ^b^	29	2.47 ± 0.04 ^b^	3.12 ± 0.02 ^c^	3.88 ± 0.07 ^c^	4.47 ± 0.13 ^c^	6.07 ± 0.33 ^c^
30PS/70RPC	2.88 ± 0.08 ^c^	29	2.93 ± 0.01 ^c^	3.01 ± 0.06 ^c^	3.63 ± 0.02 ^d^	4.46 ± 0.03 ^c^	4.48 ± 0.07 ^d^
**Mixture**	**untreated**	**Moisture content** **(g/100 g)**	**Barrel temperature**				
			**20 °C**	**80 °C**	**100 °C**	**120 °C**	**140 °C**
WPS	1.88 ± 0.05 ^a^	24	–	0.51 ± 0.05 ^a^	–	–	–
70WPS/10RPC	1.89 ± 0.15 ^a^	24	2.64 ± 0.02 ^a^	3.22 ± 0.02 ^b^	5.55 ± 0.01 ^a^	6.00 ± 0.02 ^a^	8.89 ± 0.99 ^a^
50WPS/40RPC	2.30 ± 0.04 ^b^	24	2.41 ± 0.01 ^b^	2.54 ± 0.00 ^c^	3.55 ± 0.01 ^b^	4.41 ± 0.01 ^b^	6.72 ± 0.60 ^ab^
30WPS/70RPC	2.79 ± 0.00 ^c^	24	3.05 ± 0.01 ^c^	2.96 ± 0.12 ^a^	3.65 ± 0.01 ^c^	4.28 ± 0.00 ^c^	4.79 ± 0.01 ^b^
WPS	1.88 ± 0.05 ^a^	29	–	1.42 ± 0.05 ^a^	1.12 ± 0.09 ^a^	–	–
70WPS/10RPC	1.89 ± 0.15 ^a^	29	2.64 ± 0.00 ^a^	3.65 ± 0.02 ^b^	3.56 ± 0.07 ^b^	4.16 ± 0.07 ^a^	7.91 ± 0.21 ^a^
50WPS/40RPC	2.30 ± 0.04 ^b^	29	2.44 ± 0.01 ^b^	3.00 ± 0.02 ^c^	4.33 ± 0.02 ^c^	4.45 ± 0.13 ^a^	6.49 ± 0.47 ^b^
30WPS/70RPC	2.79 ± 0.00 ^c^	29	3.19 ± 0.02 ^c^	2.93 ± 0.01 ^d^	4.13 ± 0.03 ^c^	4.13 ± 0.02 ^a^	4.67 ± 0.00 ^c^

**Table 6 polymers-13-00215-t006:** Water solubility index (WSI) of potato starch (PS) and waxy potato starch (WPS) mixtures extruded at a moisture content of 24 and 29 g/100 g and barrel temperatures of 20, 80, 100, 120 and 140 °C. Mean values with different superscript letters (a, b, c, d) within one column (starch type and moisture content) indicate significant differences (*p* > 0.05) based on a one-way analysis of variance (ANOVA). Where appropriate, mean values were compared using Tukey’s honest significance test.

Mixture	Untreated	Moisture Content (g/100 g)	Barrel Temperature				
			20 °C	80 °C	100 °C	120 °C	140 °C
PS	0.50 ± 0.00 ^a^	24	–	36.75 ± 12.37 ^a^	87.75 ± 0.35 ^a^	87.50 ± 4.24 ^a^	81.75 ± 3.18 ^a^
70PS/10RPC	8.13 ± 0.74 ^b^	24	9.20 ± 0.39 ^a^	8.28 ± 0.39 ^b^	9.16 ± 0.05 ^b^	7.76 ± 0.44 ^b^	8,11 ± 0.24 ^b^
50PS/40RPC	11.57 ± 0.08 ^c^	24	12.61 ± 0.06 ^b^	11.50 ± 0.00 ^b^	12.00 ± 0.00 ^c^	11.50 ± 0.00 ^b^	9.10 ± 0.33 ^b^
30PS/70RPC	17.39 ± 0.28 ^d^	24	20.49 ± 0.00 ^c^	19.03 ± 0.37 ^a^	18.92 ± 0.43 ^d^	14.51 ± 0.01 ^b^	14.33 ± 0.27 ^b^
PS	0.50 ± 0.00 ^a^	29	–	72.50 ± 12.73 ^a^	52.00 ± 2.12 ^a^	64.21 ± 2.54 ^a^	76.75 ± 11.67 ^a^
70PS/10RPC	8.13 ± 0.74 ^b^	29	8.06 ± 0.31 ^a^	7.33 ± 0.36 ^b^	7.15 ± 0.24 ^b^	6.59 ± 0.07 ^b^	8.15 ± 0.17 ^b^
50PS/40RPC	11.57 ± 0.08 ^c^	29	12.68 ± 0.82 ^b^	12.50 ± 0.71 ^b^	11.00 ± 0.00 ^bc^	8.75 ± 0.35 ^bc^	8.61 ± 0.21 ^b^
30PS/70RPC	17.39 ± 0.28 ^d^	29	19.04 ± 0.37 ^c^	18.42 ± 1.53 ^b^	12.83 ± 0.90 ^c^	12.66 ± 0.40 ^c^	12.52 ± 0.12 ^b^
**Mixture**	**Untreated**	**Moisture content** **(g/100 g)**	**Barrel temperature**				
			**20 °C**	**80 °C**	**100 °C**	**120 °C**	**140 °C**
WPS	0.75 ± 0.35 ^a^	24	–	82.00 ± 2.83 ^a^	–	–	–
70WPS/10RPC	6.93 ± 0.15 ^b^	24	10.08 ± 0.04 ^a^	8.07 ± 0.43 ^b^	8.48 ± 0.66 ^a^	8.06 ± 0.13 ^a^	4.78 ± 1.76 ^a^
50WPS/40RPC	11.25 ± 0.06 ^c^	24	14.19 ± 0.07 ^b^	13.08 ±0.10 ^b^	11.50 ± 0.00 ^b^	10.50 ± 0.00 ^b^	10.00 ±0.71 ^b^
30WPS/70RPC	17.95 ± 0.13 ^d^	24	18.97 ± 1.34 ^c^	20.39 ± 0.55 ^c^	17.08 ± 0.12 ^c^	14.54 ± 0.07 ^c^	12.73 ± 0.06 ^b^
WPS	0.75 ± 0.35 ^a^	29	–	42.50 ± 4.24 ^a^	57.50 ± 6.36 ^a^	–	–
70WPS/10RPC	6.93 ± 0.15 ^b^	29	7.28 ± 0.24 ^a^	3.43 ± 0.04 ^b^	4.24 ± 0.02 ^b^	5.25 ± 2.22 ^a^	5.81 ± 0.56 ^a^
50WPS/40RPC	11.25 ± 0.06 ^c^	29	14.20 ± 0.03 ^b^	11.50 ± 0.00 ^bc^	7.50 ± 0.00 ^b^	8.00 ± 0.00 ^a^	7.50 ± 0.00 ^b^
30WPS/70RPC	17.95 ± 0.13 ^d^	29	18.68 ± 0.22 ^c^	20.11 ± 0.05 ^c^	13.09 ± 0.03 ^b^	11.91 ± 0.05 ^a^	12.31 ± 0.11 ^c^

**Table 7 polymers-13-00215-t007:** Product temperatures measured for waxy potato starch (WPS) samples with 10, 40 and 70 g/100 g rapeseed press cake (RPC) extruded at a barrel temperature of 20, 80, 100, 120 and 140 °C and a moisture content of 29 g/100 g. Mean values with different superscript letters (a, b, c) within one row indicate significant differences (*p* > 0.05) based on a one-way analysis of variance (ANOVA). Mean values were compared using Tukey’s honest significance test.

	Product Temperature (°C)		
Barrel Temperature (°C)	30WPS/70RPC	50WPS/40RPC	70WPS/10RPC
**20**	52.2 ± 0.5 ^a^	55.0 ± 0.3 ^b^	62.7 ± 1.6 ^c^
**80**	90.2 ± 0.4 ^a^	82.7 ± 1.1 ^b^	88.2 ± 1.3 ^c^
**100**	100.0 ± 0.2 ^a^	96.7 ± 0.1 ^b^	95.3 ± 0.8 ^c^
**120**	108.8 ± 0.5 ^a^	107.5 ± 0.6 ^b^	104.8 ± 0.4 ^c^
**140**	123.4 ± 0.4 ^a^	125.7 ± 1.0 ^b^	124.2 ± 1.6 ^c^

**Table 8 polymers-13-00215-t008:** Extruder system parameters used for starch based extrudates containing waxy potato starch (WPS) mixed with rapeseed press cake (RPC) extruded at a barrel temperature of 20, 80, 100, 120 and 140 °C. SME = specific mechanical energy. Mean values with different superscript letters (a, b, c, d, e) within one column (mixture and moisture content) indicate significant differences (*p* > 0.05) based on a one-way analysis of variance (ANOVA). Mean values were compared using Tukey’s honest significance test.

Mixture	Moisture Content (g/100 g)	Barrel Temperature (°C)	PS		WPS	
			SME (kWh/kg)	Pressure (bar)	SME (kWh/kg)	Pressure (bar)
30starch/70RPC	24	20	0.11 ± 0.01 ^a^	6.94 ± 0.73 ^a^	0.12 ± 0.00 ^a^	12.90 ± 1.87 ^a^
		80	0.09 ± 0.00 ^a^	10.63 ± 0.24 ^b^	0.11 ± 0.00 ^a^	6.49 ± 0.41 ^b^
		100	0.08 ± 0.00 ^a^	5.42 ± 1.48 ^a^	0.09 ± 0.00 ^a^	5.98 ± 0.50 ^b^
		120	0.08 ± 0.00 ^a^	2.39 ± 1.35 ^c^	0.08 ± 0.00 ^a^	5.04 ± 0.31 ^b^
		140	0.07 ± 0.00 ^a^	1.74 ± 1.03 ^c^	0.08 ± 0.00 ^a^	3.51 ± 0.46 ^c^
	29	20	0.10 ± 0.01 ^a^	6.12 ± 1.26 ^a^	0.10 ± 0.00 ^a^	13.75 ± 0.58 ^a^
		80	0.08 ± 0.00 ^a^	9.49 ± 0.82 ^b^	0.08 ± 0.00 ^a^	6.18 ± 0.67 ^b^
		100	0.07 ± 0.00 ^a^	6.47 ± 1.14 ^a^	0.08 ± 0.00 ^a^	4.53 ± 0.43 ^c^
		120	0.07 ± 0.00 ^a^	4.13 ± 0.48 ^c^	0.07 ± 0.00 ^a^	2.83 ± 0.71 ^d^
		140	0.07 ± 0.00 ^a^	0.62 ± 0.91 ^d^	0.07 ± 0.00 ^a^	0.60 ± 0.07 ^e^
50starch/40RPC	24	20	0.10 ± 0.01 ^a^	13.43 ± 2.39 ^a^	0.10 ± 0.01 ^a^	8.11 ± 1.24 ^a^
		80	0.08 ± 0.00 ^a^	6.69 ± 0.78 ^b^	0.09 ± 0.01 ^a^	4.84 ± 2.57 ^b^
		100	0.08 ± 0.00 ^a^	2.78 ± 0.56 ^c^	0.08 ± 0.01 ^a^	4.87 ± 2.22 ^b^
		120	0.07 ± 0.00 ^a^	0.68 ± 1.38 ^d^	0.08 ± 0.00 ^a^	2.63 ± 1.90 ^c^
		140	0.07 ± 0.00 ^a^	0.25 ± 0.94 ^d^	0.07 ± 0.00 ^a^	0.26 ± 0.04 ^d^
	29	20	0.09 ± 0.00 ^a^	10.92 ± 1.10 ^a^	0.09 ± 0.00 ^a^	10.57 ± 0.40 ^a^
		80	0.07 ± 0.00 ^a^	7.08 ± 0.46 ^b^	0.07 ± 0.00 ^a^	8.82 ± 1.64 ^b^
		100	0.07 ± 0.00 ^a^	2.95 ± 0.70 ^c^	0.07 ± 0.00 ^a^	3.67 ± 0.35 ^c^
		120	0.06 ± 0.00 ^a^	1.28 ± 0.45 ^d^	0.07 ± 0.00 ^a^	0.66 ± 0.14 ^d^
		140	0.06 ± 0.00 ^a^	1.08 ± 0.56 ^d^	0.06 ± 0.00 ^a^	0.70 ± 0.17 ^d^
70starch/10RPC	24	20	0.11 ± 0.00 ^a^	16.86 ± 1.72 ^a^	0.12 ± 0.00 ^a^	40.51 ± 5.28 ^a^
		80	0.09 ± 0.00 ^a^	8.67 ± 1.98 ^b^	0.10 ± 0.00 ^ab^	32.10 ± 6.25 ^b^
		100	0.08 ± 0.00 ^a^	4.53 ± 0.23 ^c^	0.09 ± 0.01 ^ab^	25.08 ± 5.48 ^c^
		120	0.07 ± 0.00 ^a^	4.12 ± 5.21 ^c^	0.08 ± 0.01 ^ab^	14.04 ± 3.91 ^d^
		140	0.07 ± 0.00 ^a^	7.34 ± 4.75 ^b^	0.07 ± 0.00 ^b^	11.28 ± 2.71 ^d^
	29	20	0.07 ± 0.01 ^a^	10.81 ± 2.76 ^a^	0.11 ± 0.00 ^a^	31.42 ± 6.99 ^a^
		80	0.09 ± 0.00 ^a^	14.41 ± 5.51 ^b^	0.09 ± 0.00 ^a^	16.33 ± 5.85 ^b^
		100	0.08 ± 0.00 ^a^	8.14 ± 4.67 ^c^	0.07 ± 0.00 ^a^	8.52 ± 1.53 ^c^
		120	0.07 ± 0.00 ^a^	4.91 ± 3.80 ^d^	0.07 ± 0.00 ^a^	6.68 ± 1.35 ^c^
		140	0.07 ± 0.00 ^a^	5.84 ± 3.48 ^d^	0.07 ± 0.00 ^a^	2.11 ± 1.37 ^d^

## Data Availability

The data presented in this study are available on request from the corresponding author.

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
