# Peer review of "Linking Expansion Behaviour of Extruded Potato Starch/Rapeseed Press Cake Blends to Rheological and Technofunctional Properties"

_polymers, 2021, doi:10.3390/polym13020215_

Round 1

Reviewer 1 Report

The article is very interesting one. The methodology, the data presentation is well written and defined. I have only some minor remarks:

  • to the introduction part the authors should underline better the novelty of their research;
  • the authors underline in the introduction part some antinutritional aspects of rapeseed products. My question is if the raw materials used by them are safe or not to be used?
  • the conclusions are too long, difficult to read. It must be presented more shorter. 

Reviewer 2 Report

Review on manuscript polymers-1068676:

Linking expansion behaviour of extruded potato starch/rape-seed press cake blends to rheological and technofunctional properties

by Anna Martin, Raffael Osen, Heike Petra Karbstein and M. Azad Emin

submitted to Polymers

In the manuscript submitted for comments the Authors studied the impact of the rape-seed press cake content, starch type and extrusion temperature on the expansion, physical quality and extruder response of extruded starch/ rape-seed press cake blends.

Generally the manuscript is prepared correctly, so after minor revision could be consider for publication in the Polymers journal.

Detailed recommendation:

supplementary materials – the abbreviations list should be included in the main manuscript,

introduction/aim of the study/conclusion – the authors should show in a more expressive way that their research is also of an application nature, and the extrudates obtained based on starch and rape-seed press cake can be used as e.g. snacks with increased nutritional value,

Tables – the order in which the tables appear in the manuscript is inconsistent with their numbering,

lines 145-148 and 318, 325 – are WAI and WSI really dimensionless quantities? most often WAI is expressed in g/g and WSI in%,

Table 1 – lack of unit for DM content,

line 178 – Dry matter content …. in what units?

lines 229-240 – the authors give the content and discuss the influence of the dietary fiber fraction, although the methodology does not state that this component was determined,

lines 494, 497 – Italic style should be used,

lines 473, 513 and 543 – the abbreviation should be used,

line 562 – incomplete citation,

line 565 – journal name should be written with capital letters.
